# Combined Metabolipidomic and Machine Learning Approach in a Rat Model of Stroke Reveals a Deleterious Impact of Brain Injury on Heart Metabolism

**DOI:** 10.3390/ijms241512000

**Published:** 2023-07-26

**Authors:** Xavier Dieu, Sophie Tamareille, Aglae Herbreteau, Lucie Lebeau, Juan Manuel Chao De La Barca, Floris Chabrun, Pascal Reynier, Delphine Mirebeau-Prunier, Fabrice Prunier

**Affiliations:** 1MITOVASC, SFR ICAT, CNRS, INSERM, Université d’Angers, F-49000 Angers, France; sophie.tamareille@univ-angers.fr (S.T.); aglae.herbreteau@univ-angers.fr (A.H.); lucie.lebeau@univ-angers.fr (L.L.); JMchaodelabarca@chu-angers.fr (J.M.C.D.L.B.); floris.chabrun@chu-angers.fr (F.C.); pareynier@chu-angers.fr (P.R.); deprunier@chu-angers.fr (D.M.-P.); faprunier@chu-angers.fr (F.P.); 2Service de Biochimie et Biologie Moléculaire, CHU Angers, F-49000 Angers, France; 3Service de Cardiologie, CHU Angers, F-49000 Angers, France

**Keywords:** stroke heart syndrome, metabolomics, machine learning

## Abstract

Cardiac complications are frequently found following a stroke in humans whose pathophysiological mechanism remains poorly understood. We used machine learning to analyse a large set of data from a metabolipidomic study assaying 630 metabolites in a rat stroke model to investigate metabolic changes affecting the heart within 72 h after a stroke. Twelve rats undergoing a stroke and 28 rats undergoing the sham procedure were investigated. A plasmatic signature consistent with the literature with notable lipid metabolism remodelling was identified. The post-stroke heart showed a discriminant metabolic signature, in comparison to the sham controls, involving increased collagen turnover, increased arginase activity with decreased nitric oxide synthase activity as well as an altered amino acid metabolism (including serine, asparagine, lysine and glycine). In conclusion, these results demonstrate that brain injury induces a metabolic remodelling in the heart potentially involved in the pathophysiology of stroke heart syndrome.

## 1. Introduction

Stroke heart syndrome designates the diverse cardiac complications that may follow a stroke. It is estimated that following a stroke, up to 20% of patients present at least one cardiac complication [1]. Myocardial injury or infarction, takotsubo syndrome, ventricular dysfunction or arrhythmia, notably atrial fibrillation, are the most common manifestations of this syndrome [2].

Several mechanisms may participate in this secondary cardiac involvement, including neuro-cardiac and neuro-hormonal regulation, especially with the central role of catecholamines, immune and inflammatory responses, notably with the involvement of immune cells such as macrophage and the key role of cytokines such as IL-1, and, finally, the role of the gut–heart axis through gut dysbiosis [2,3].

In this study, we hypothesized that metabolic impairment may also be involved in the deleterious impact of brain injury to the heart. Indeed, metabolic changes have been highlighted in patients following a stroke, such as those involving amino acids [4]. Compared to healthy controls, the blood of stroke patients consistently showed decreased concentrations of glycine, valine, isoleucine, leucine, lysine, citrate, alanine and serine and increased concentrations of glutamate and lactate. The metabolic pathways most often found altered after a stroke involve glycine, serine, threonine, valine, leucine, isoleucine, glutathione and folate metabolisms [5]. Lipids are also consistently found to be altered following a stroke, notably blood levels of lysophosphatidylcholines and phosphatidylcholines, which several studies have found to be decreased and linked to inflammatory responses. Levels of sphinganine, a ceramide precursor, have been shown to be increased in stroke patients, while ceramides have been shown to be linked with higher risk of ischemic stroke [4]. Overall, these changes in metabolite concentrations in stroke patients are involved in energy metabolism, oxidative stress, excitotoxicity and inflammation [4,5,6].

Metabolomics is a powerful tool allowing for the quantification of an increasing number of metabolites, which in turn allows us to capture and test the predictive power of the metabolic signatures associated with pathological conditions. It has been successfully used to understand metabolic pathways involved in the pathophysiology of stroke, both in human and animal models [6]. The use of targeted metabolomics assays such as the one used in our study allows the precise and rapid quantification of a finite and predefined number of metabolites compared to untargeted methods. These methods are highly standardized, thanks to the use of internal controls (low, medium and high levels), as well as calibration curves. The results are therefore highly comparable between user laboratories, highly suitable for clinical and preclinical research. It was used to unravel new pathophysiological mechanisms in multiple pathologies such as arterial hypertension [7], sickle cell disease [8] or myocardial infarction and remote ischemic conditioning [9] where results were demonstrated to be well correlated with untargeted metabolomics [10]. To date, to our knowledge, metabolomics have never been used to investigate stroke heart syndrome in patients or animal models, specifically in the search for the metabolic impact of stroke on the heart metabolism.

Continuous improvements of metabolomic pipelines make it possible to study an increasing number of metabolites, which contribute to generate larger and more complex datasets. Traditional statistical approaches are themselves limited in this setting and the use of machine learning is thought to facilitate better performance in this context [11]. Indeed, machine learning approaches allow us to model nonlinear data representation as well as large and heterogeneous data sets [12].

Comparing a rat stroke model with controls, we investigated whether a machine learning approach could unravel a metabolomic signature specifically affecting the heart in comparison with the blood signature.

## 2. Results

Of the 60 rats initially included in the study (30 rats undergoing the sham procedure and 30 rats undergoing the stroke procedure), 10 did not survive the experimental procedure (two in the sham and eight in the stroke group). Furthermore, we eliminated 10 rats from the stroke group that had a percentage of brain necrosis below 10% based on post-mortem brain histological sections, guaranteeing a significant stroke was generated. The remaining 40 rats were thus explored, 28 of which were exposed to the sham procedure, whereas 12 had a significant stroke with brain infarction. Mean ± standard deviation for the percentage of necrosis evaluated on brain histological sections was 25.9 ± 10.4%. After applying quality controls and filtering as described in the methods section, we accurately measured 400 metabolites in rat plasma and 154 metabolites in heart extracts (full metabolites list in Appendix A).

On plasma samples, principal component analysis (PCA) displayed no spontaneous clustering according to the stroke or sham procedure, and it did not reveal any outliers. On the heart samples, PCA displayed no spontaneous clustering but it highlighted one outlier sample that was removed from subsequent analyses.

Random stratified sampling was carried out: for the plasma dataset, the training set comprised 28 rats and the remaining 12 rats were kept for the test set, whereas for the heart dataset the training set comprised 28 rats and the remaining 11 rats were kept for the test set. A machine learning approach was used, including feature selection, testing and fine-tuning of multiple classical machine learning models for the classification of rats subjected or not subjected to stroke based on plasma and heart samples. The whole machine learning pipeline was reiterated 30 times to obtain more stable and robust results. A global overview of the study design is described in Figure 1.

### 2.1. Metabolomic Signature in Plasma

On plasma samples, the best obtained model was a support vector machine that achieved a median AUC-ROC on the test set of 0.91 (±0.1) on the 30 different samples of the plasma dataset into training and test data (Figure 2). Twenty metabolites contributing to the stroke prediction were identified as the most consistently selected for achieving this performance (Figure 3, Table 1). This signature highlighted alterations in the concentration of six different phosphatidylcholines and lyso-phosphatidylcholines and seven sphingolipids (five different ceramides and hexosylceramides as well as two sphingomyelins). Some sphingomyelins (C16:0 and C14:1), ceramides (d18:1/16:0) and hexosylceramides (d18:1/16:0, d18:1/20:0) had higher concentrations in the stroke group, whereas some ceramides were found decreased in this group. Phosphatidylcholines (aa C42:2, aa C42:5, aa C40:2, ae C38:2) and lyso-phosphatidylcholines (a C26:1, a C28:1) showed decreased concentrations. In addition, sum/ratios univariate analysis found that the ratio of sphingomyelin to phosphatidylcholines was increased in stroke group.

Regarding the amino acids component of the stroke signature, we found alterations in the concentrations of (by decreasing order of importance) trans-4-hydroxyproline, citrulline, lysine and ornithine. Sum/ratios univariate analysis found that the ratio of proline to citrulline (indicator of the relative arginase activity compared to NO synthase) was increased in the stroke group, the activity of NO-synthase (citrulline/arginine) was decreased in the stroke group, and the global arginine bioavailability ratio was augmented in this group.

Other metabolites, such as 3-indolepropionic acid (3-IPA), cholic and desoxycholic acid (Figure 3), were also impacted by stroke. They were all decreased in the plasma of the rats in the group that had strokes compared to the sham group. Finally, the ratio of dihexosylceramides to ceramides was found diminished in the stroke group.

### 2.2. Cardiac Metabolomic Signature

On heart samples, one rat was detected as an outlier with PCA and was removed from subsequent analysis. The best obtained model was also a support vector machine that achieved a median AUC-ROC on the test set of 0.89 (±0.16) on the 30 different train/test sampling of the heart data (Figure 2).

Among the 154 metabolites accurately measured in heart samples, seven important metabolites were identified as the most consistently selected for our models’ performances (Figure 4, Table 2).

No particular lipid signature was found in this tissue. The amino acids (by decreasing order of importance) trans-4-hydroxyproline, lysine, serine and asparagine were found consistently increased in the left ventricles of rats after stroke, compared to the sham group. Sum/ratios univariate analysis found that the ratio of proline to citrulline was augmented in the stroke group, the activity of NO-synthase (citrulline/arginine) was diminished in the stroke group, the glycine synthesis ratio (glycine/serine) was diminished in the stroke group and the sum of ketogenic amino acids was augmented in the stroke group.

Three other metabolites were found with altered concentrations: 5-aminovaleric acid, cholic acid and betaine (Figure 4). These three metabolites were decreased in the stroke group compared to the sham group (Figure 5).

### 2.3. Common Metabolomic Signature in Heart and Blood

Three metabolites were found in common between the plasma and heart samples. Two metabolites had higher concentrations in the stroke group compared to the sham group, including lysine and trans-4-hydroxyproline, both of which had a log2 fold change (stroke/sham) of 0.41. The third metabolite, cholic acid, was found decreased in the stroke group with an average log2 fold change (stroke/sham) of −1.1 (Figure 5). Ratios of proline to citrulline were augmented in both samples of the stroke group, and the NO-synthase activity (citrulline/arginine) was diminished in both samples of the stroke group.

We used metabolite set enrichment analysis to interpret these metabolomic signatures in the heart and plasma. We found significant enrichment of the arginine and proline metabolism as well as arginine biosynthesis, both of which were similar to our findings using metabolic ratios (Figure 6).

## 3. Discussion

The main objective of our study was to investigate whether a biological signature could be detected in the hearts of rats subjected to stroke. With a combined targeted metabolomics and machine learning approach, we detected such a signature. This biological signature appearing in the heart, which involves seven metabolites with modified concentrations, is much more modest than the signature appearing in the blood, which comprises 20 discriminant metabolites. It nevertheless attests to the indirect deleterious impact of stroke on the heart observed in clinical settings. This signature constitutes an opportunity to explore the mechanisms involved in this secondary heart damage.

### 3.1. Phospholipids Signature

An important lipids signature was found in the plasma, but not in the heart of rats in the days following stroke. Phosphatidylcholines (PC) have a choline head group, a glycerophosphoric acid and two fatty acids of different sizes. They are particularly present in cell membranes and are involved in multiple processes such as inflammation. Lysophosphatidylcholines (LPC) are derived from PC thanks to the hydrolytic action of phospholipases A2 or A1. LPCs are also known for their role in myelin sheath phagocytosis through the recruitment of macrophages and microglia [13]. Multiple studies have found them to be diminished in the plasma of patients after a stroke. In a lipidomic study on the plasma of 35 patients following a stroke, 13 lipids were found altered in comparison to the 21 healthy controls. Four LPCs were found up-regulated in patients, a potential sign of phospholipase A2 activation [14]. Another study comparing 66 patients after ischemic stroke versus 63 controls found a decrease in PC and LPC levels after stroke [15]. Previous rat models of stroke have found decreased PC and LPC in serum [5,16,17]. Accordingly, we also found PC and LPC levels diminished in our rat model, which can be attributed, at least partly, to the inflammatory processes found following a stroke. A publication studying a rat model of ischemic stroke also found a pattern of decreased PC and increased LPC levels in brain sections following stroke [18].

### 3.2. Sphingolipids Signature

Sphingolipids are essential components of cellular membranes, which also regulate diverse cell functions. Among sphingolipids, ceramides occupy a central position and are well known as lipid mediators involved in cell death, differentiation, senescence and autophagy [19]. Ceramides also have a role in plaque formation and thrombosis [20]. They have been found increased in the plasma of patients in an untargeted metabolomics study comparing 42 patients in the week after an ischemic stroke with 30 controls [21]. Ceramides are, furthermore, useful in risk prediction, as showcased in a study on 1767 patients followed for 7 years; here, plasma ceramide concentrations were found positively associated with cardiovascular outcomes [22]. In a study of the plasma of patients following a stroke, glucosylceramides were found decreased and were a strong predictor of stroke status [14]. We also found that rats in the stroke group had a diminished proportion of dihexosylceramides (Hex2Cer) relative to the hexosylceramide (HexCer) pool. This result supports the observation that the synthesis of hexosylceramides is a way to control the pool of ceramides [23].

Alongside all of the above, we found an augmented ratio of sphingomyelins to phosphatidylcholines in the plasma of rats after a stroke. It represents sphingomyelin synthase activity through sphingomyelin synthesis from phosphatidylcholines. Sphingomyelin synthase generates sphingomyelin from ceramide by the transfer of phosphocholine from phosphatidylcholine with the generation of diacylglycerol. Commonly found in plasma membranes and in myelin sheath, sphingomyelins are structured around a phosphocholine head group, a sphingosine and a fatty acid of variable length. In conjunction with ceramides, sphingomyelin have also been found to be increased in the plasma of patients following an ischemic stroke [21]. In our study, we found two sphingomyelins significantly augmented in plasma samples of rats after stroke, in parallel with higher activity of sphingomyelin synthase. Interestingly, on the one hand, mice KO for sphingomyelin synthase had inhibited atherosclerosis as well as decreased the inflammatory response. By contrast, overexpression of sphingomyelin synthase promoted atherosclerosis over weeks through lowering non-HDL lipoprotein retention/aggregation in plaque [19]. On the other hand, ceramides, the precursor of sphingomyelin, have an increased synthesis thanks to the stimulating role of inflammatory cytokines such as TNFα and IL-1β [23]. The inflammatory responses elicited by a stroke therefore contribute to increased ceramides and sphingomyelin production, which have pro-apoptotic, autophagic and atherosclerotic effects. These shingolipid alterations may contribute to cardiac complications by accelerating atherosclerosis and plaque rupture as well as pro-thrombotic mechanisms found in stroke heart syndrome.

Interestingly, we found this lipid signature in plasma, but not in heart samples of rats after a stroke. This sphingolipids signature is coherent with the other studies in humans, but was not as extensively described in rat models, which validates the relevance of our approach. It is likely to result from brain infarction and subsequent releases of these lipids present in high quantities in the brain.

### 3.3. Amino Acid Signature

Trans-4-hydroxyproline levels were augmented in both the plasma and hearts of rats subjected to stroke. Hydroxyproline, generated through collagen hydrolysis, is a marker of collagen turnover, especially from skeletal muscle and bone mass. In heart samples, it means that a remodelling activity may be present in the heart in the days immediately following a stroke, a phenomenon that could be linked to cardiac complications [2]. This elevation of trans-4-hydroxyproline was also found in the plasma, where it could also be due to the brain damage caused by a stroke. This finding has not been described in previous rat models of stroke.

The ratio of proline to citrulline (proline/citrulline) was significantly higher in the stroke group compared to the sham group in both plasma and heart samples. This ratio is an indicator of the conversion of ornithine to proline by ornithine aminotransferase and pyrroline-5-carboxylate reductase and serves as an indirect indicator of the relative arginase activity that competes with NOS activity. Inflammatory factors and atherothrombosis mediators, such as oxidized low-density lipoprotein and thrombin, can activate arginases. It also results in depriving arginine from the NOS pathway, decreasing NO bioavailability, which has been linked to inflammation, endothelial dysfunction as well as cardiovascular disorders, including atherosclerosis [24]. The decreased NO bioavailability could therefore be an additional mechanism contributing to the local cardiac complications found in this syndrome such as myocardial lesions to infarctions due to endothelial dysfunction and/or atherosclerotic plaque destabilization or rupture. In parallel, we found that arginine bioavailability was increased in the plasma of rats from the stroke group. This is surprising since lower arginine bioavailability has been found to correlate with endothelial dysfunction and cardiovascular mortality in a longitudinal study involving 2236 patients followed for coronary artery disease for eight years [25]. Enriched metabolic pathways also confirmed the importance of these same significant pathways, namely arginine and proline metabolism and arginine biosynthesis. This highlights the critical role of NO generation and regulation in both plasma and heart samples that is impaired after stroke.

The serine to glycine ratio is an indicator of mitochondrial serine hydroxymethyltransferase (SHMT) activity converting serine to glycine. It was found to be diminished in heart samples of rats after a stroke in conjunction with an elevated serine level. Serine and glycine are non-essential amino acids that contribute to one carbon metabolism, with roles in nucleotide synthesis, the folate cycle and responses to oxidative stress through glutathione synthesis. This elevation of serine may hint at a lower utilization of serine to fuel one carbon metabolism, notably glycine synthesis, leading to a reduction in cells’ ability to sustain oxidative stress damage [26]. Simultaneously, we found elevated asparagine levels in the hearts of rats from the stroke group. Asparagine is synthesized from glutamine, and has been linked with amino acid uptake, notably the uptake of serine [27], hence providing another explanation for high serine levels in the same rats. Asparagine production is found to be correlated with macrophage activation [28], which points to the role of the systemic inflammation found following a stroke as an explanation for asparagine augmentation.

### 3.4. Other Metabolites

Betaine, also known as trimethylglycine, was found diminished in the heart samples of rats following a stroke. It has a neuroprotective role, preserves myocardial function and prevents liver diseases mainly through its anti-oxidant and anti-inflammatory properties. These properties come from its ability to protect cells from osmotic stress, being a methyl donor in the conversion of homocysteine to methionine, and inhibiting NF-κB and NLRP3 inflammasome activity [29,30]. The depletion of betaine in the heart following a stroke highlights the vulnerability of the myocardium to further inflammatory processes and to oxidative stress. Low levels of betaine correlate with increased risk of secondary heart failure and acute myocardial infarction [31].

Cholic and desoxycholic acids are bile acids that were found to be diminished in the plasma and heart samples of rats belonging to the stroke group. They are derived from cholesterol and have gained attention in recent years for roles as vasoactive ligands involved in vascular tone and myocardial contractility on top of more general roles in energy metabolism, cell proliferation and immunomodulation. Cholic acid has been described to have an arrythmogenic effect [32,33]. Elevated levels of bile acids have been linked with cardiac dysfunction through the inhibition of fatty acid β-oxidation. In view of the data in the literature and the decrease in bile acids in our experiment, there is no evidence to date that these metabolites are involved in stroke heart syndrome.

Two microbiota-derived metabolites were found to be altered in rats after a stroke. First, 5-aminovaleric acid (5-AVA) was elevated in heart samples of rats following a stroke, which might reflect alterations in the gut–heart axis [34], and 3-indolepropionic acid (3-IPA), a microbiota-derived metabolite of tryptophan, was found to be diminished in the plasma of rats belonging to the stroke group, which might reflect the impact of gut dysbiosis found in stroke heart syndrome [3]. This metabolite has multiple roles, notably limiting inflammation, diminishing lipid peroxidation and anti-oxidant function. Circulating 3-IPA levels were found to be decreased in cardiovascular pathologies such as atherosclerosis [35]. This decrease could also be linked to inflammation or oxidative stress states during stroke.

### 3.5. Limits

Our study has some limitations. It is based on a rat model, which means that some of our observations may not be applicable to humans. Also, the use of targeted metabolomics, while allowing precise measurements, limits our analysis to a predefined set of metabolites, excluding metabolites that are potentially significant in heart injury. Finally, several rats did not survive the experimental procedure, which led to a lower number of rats being in the stroke group than in the control group.

## 4. Materials and Methods

### 4.1. Animals

All the experiments were performed in accordance with the European Community Guiding Principles for the care and use of animals (Directive 2010/63/UE; Décret n°2013-118). The study protocol was approved by the regional ethics committee (Comité d’Ethique en Experimentation Animale des Pays de la Loire) and by the French Ministry of Higher Education and Research (agreement APAFIS#27253-2020091713561456 v7). We used Sprague–Dawley male rats weighing 300–350 g (Envigo, Gannat, France). They were allowed to acclimatize for at least seven days before entering the experiment. The temperature (22 ± 2 °C) and humidity (55 ± 20%) were controlled with a 12/12 h light/dark cycle. The access to pelleted food and water was ad libitum. Rats were pre-emptively allocated to two groups: 30 were allocated to the sham group and 30 to the stroke group.

### 4.2. Transient Middle Cerebral Artery Occlusion

The intraluminal filament model of focal ischemia was used. Rats were anaesthetised by intraperitoneal injection of sodium pentobarbital (60 mg/Kg) and analgesia was obtained with subcutaneous administration of buprenorphine at 0.05 mg/kg. The body temperature was monitored using a rectal probe and maintained at 37 ± 0.5 °C throughout the procedure using a heating mat. After a midline neck incision, a standardized silicone-coated 6-0 nylon monofilament (Doccol Corporation, Sharon, MA, USA) was inserted into the right internal carotid artery and advanced to occlude the origin of the middle cerebral artery. The filament was kept in place for 90 min, and then withdrawn allowing reperfusion. The sham animals underwent the same procedure, but without the insertion of the filament to the right internal carotid artery. Rats were euthanized at 72 h after the experimental stroke procedure. Blood from vena cava and left ventricle samples were collected for subsequent analyses.

### 4.3. Triphenyl Tetrazolium Chloride (TTC) Staining

TTC staining was used to assess histological necrosis. The brains were quickly extracted, and sectioned at 1.5 mm intervals with a brain slicer matrix. The brain slices were stained with 1% 2,3,5-triphenyltetrazolium chloride (TTC) for 8 min and fixed in formalin. They were then photographed, and the infarcted areas were measured blindly using ImageJ software (v1.53p). Brain infarction was expressed as a percentage of total brain area. To guarantee a sufficient biological stroke effect, we included rats for statistical processing only if they had brain infarctions superior to 10% based on post-mortem brain histological sections.

### 4.4. Analytical Workflow

Plasma samples were prepared from the blood and stored at −80 °C. The left ventricle samples were immediately plunged into liquid nitrogen before their conservation at −80 °C until the extraction of metabolites. Before extraction, the samples were weighed using an XA105DU analytical balance (Mettler Toledo, Viroflay, France) with an accuracy of 0.01 mg and the weight of the samples was used for in-between sample normalization. Tissue samples were collected in pre-cooled (dry ice) 2.0 mL homogenization Precellys tubes prefilled with 1.4 mm diameter ceramic beads and 3 μL/mg cold methanol (Zukunft et al., 2018). Tissues were homogenized by two grinding cycles, each at 6600 rpm for 20 s, spaced 20 s apart, using a Precellys homogenizer (Bertin Technologies, Montigny-le-Bretonneux, France) kept at +4 °C. The supernatant was recovered after centrifuging the homogenate and kept at −80 °C until mass spectrometric analysis. We used a targeted quantitative metabolomic approach using the Biocrates^®^ MxP^®^ Quant 500 kit (Biocrates Life sciences AG, Innsbruck, Austria). This kit allows for the quantification of up to 630 metabolites and the computation of 234 ratios or sums of metabolites indicative of various metabolic processes using mass spectrometry (QTRAP 5500, SCIEX, Villebon-sur-Yvette, France). These ratios and sums of metabolites are indicators of a wide range of metabolic functions and pathways. Carnitine, acylcarnitines, lipids and hexoses were analysed by flow injection analysis coupled with tandem mass spectrometry (FIA-MS/MS), whereas amino acids and biogenic amines were quantified using liquid chromatography coupled with tandem mass spectrometry (LC–MS/MS). The distribution of samples of the different rats was randomized according to the experimental procedure (with or without stroke) over the plate to prevent batch-effect variations: one run was used for analysing plasma samples and a second one was used to study heart samples. Peaks were integrated in order to retrieve raw data as a matrix including the measurements of the 630 metabolites, in addition to pool, calibration and quality control samples. Three quality controls (QCs) composed of three concentrations of human plasma samples, i.e., low (QC1), medium (QC2) and high (QC3), were used to evaluate the performance of the analytical assay. A seven-point serial dilution of calibrators was used to generate calibration curves for the quantification of amino acids and biogenic amines. We applied a filtering step designed to remove unusable data: metabolites with more than 30% of values lying outside the limits of quantification were removed if no statistically significant association between the out of bound values of the metabolite and the class (sham or stroke) was found with a Chi-squared test (i.e., a metabolite is only quantifiable in one group and not in the other). We thus obtained two separate datasets (one for plasma and the other for heart samples) that were independently processed through the statistical workflow.

### 4.5. Statistical Workflow

Data pre-processing including fold-change calculation for each compound, matrix standardisation with mean centring and variance scaling, was carried out. Univariate analyses of all metabolites and ratios or sums of metabolites were achieved by using Mann–Whitney U tests with Benjamini–Hochberg correction. Tests were considered statistically significant if the corrected p-value was below 0.05. Multivariate unsupervised Principal Components Analysis (PCA) was used to identify outliers as well as spontaneous clusters. Then, a machine learning pipeline was used to analyse each dataset (plasma and heart samples) separately. We used random stratified sampling to split each dataset into a training set (70% of the initial samples) and a test set (approximately 30% of the initial samples, in order to retain at least 12 samples in the test set). The stratified sampling allowed us to respect the original proportion of sham and stroke samples in both the train and test datasets. We use the train set to perform a feature selection of relevant variables while discarding non-informative ones, using the BorutaPy algorithm [36]. Features selected were then used to train multiple machine learning models including regularized logistic regression, support vector machines (with either linear or nonlinear kernels) and random forest. Each model had its hyperparameters fine-tuned on 10 resamplings of the training set into a train and a validation set with a bootstrap without replacement strategy. The best model performance was then assessed on the test set with its AUC-ROC (Area under Receiver Operating Characteristic Curve). This whole process (data partitioning, feature selection, model selection and testing) was repeated on 30 different random stratified samplings of the initial data into different train and test datasets to average the results and metrics of the best performing model, allowing for more robust results. Average feature importance was computed using the SHAP values [37] of each selected metabolite through the feature selection process, across the 30 iterations. In order to better highlight the most important metabolites for a correct prediction, models whose AUC-ROC belonged to the lower 25 percentile of all models were excluded. Then, metabolites that were selected in more than half of the iterations of the data partitioning/models remaining were considered as significantly contributing to the best model performance across varying data splits. Finally, over representation analysis was conducted with top metabolites with the MetaboAnalyst 5.0 web application (https://www.metaboanalyst.ca/, accessed on 8 september 2022) [38] to identify significantly enriched metabolic pathways in plasma and heart samples. We only retained metabolites that were selected at least 3 times with our feature selection methods.

### 4.6. Computational Tools

All data processing was performed using Python (3.7.7) and Python packages including: pandas (1.0.5), scikit-learn (0.23.1), tensorflow (2.2.0), BorutaPy (0.3), SHAP (0.40.0), seaborn (0.10.1), matplotlib_venn_wordcloud (0.2.5) and matplotlib (3.3.0).

## 5. Conclusions

This study identified metabolites that have already been shown to change following a stroke in the blood of patients and animal models, such as PC, LPC and ceramides. It also revealed metabolites newly involved in the plasma in the context of stroke, such as 3 indolepropionic acid and cholic acid. Most importantly, this study revealed for the first time a metabolic signature appearing specifically in the heart after stroke, involving the following seven metabolites: lysine, serine, asparagine, trans-4-hydroxyproline, cholic acid, 5-aminovaleric acid and betaine. This signature suggests an increase in collagen turnover, an increase in arginase activity with a decrease in NO production and an alteration in the metabolism of serine, asparagine, lysine and glycine. These metabolic changes provide evidence that brain injury induces metabolic changes in the heart that are likely to be involved in the pathophysiology of stroke-related cardiac damage. Additionally, these metabolite changes need to be further studied in patients after stroke as they may be potential biomarkers which could correlate with the apparition of potential secondary cardiac complications. In the future, altered pathways in heart after stroke could be interesting to explore in order to find potential therapeutic targets which could prevent these complications.

## Figures and Tables

**Figure 1 ijms-24-12000-f001:**
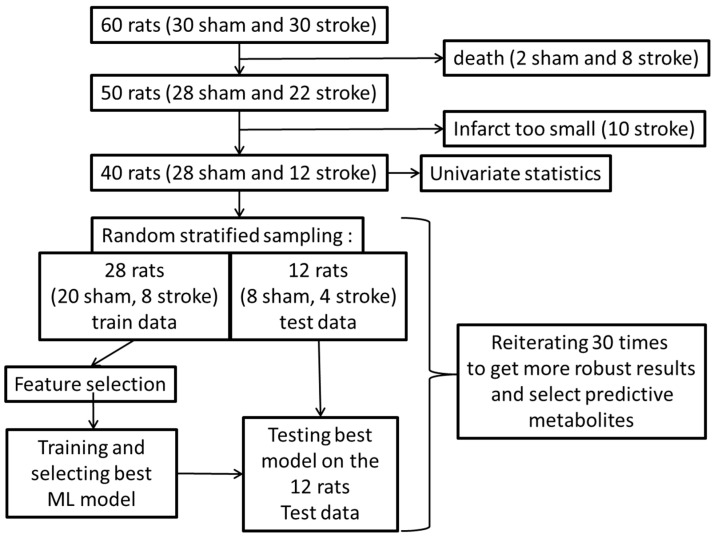
Global overview of the study flowchart and statistical analyses. ML: machine learning.

**Figure 2 ijms-24-12000-f002:**
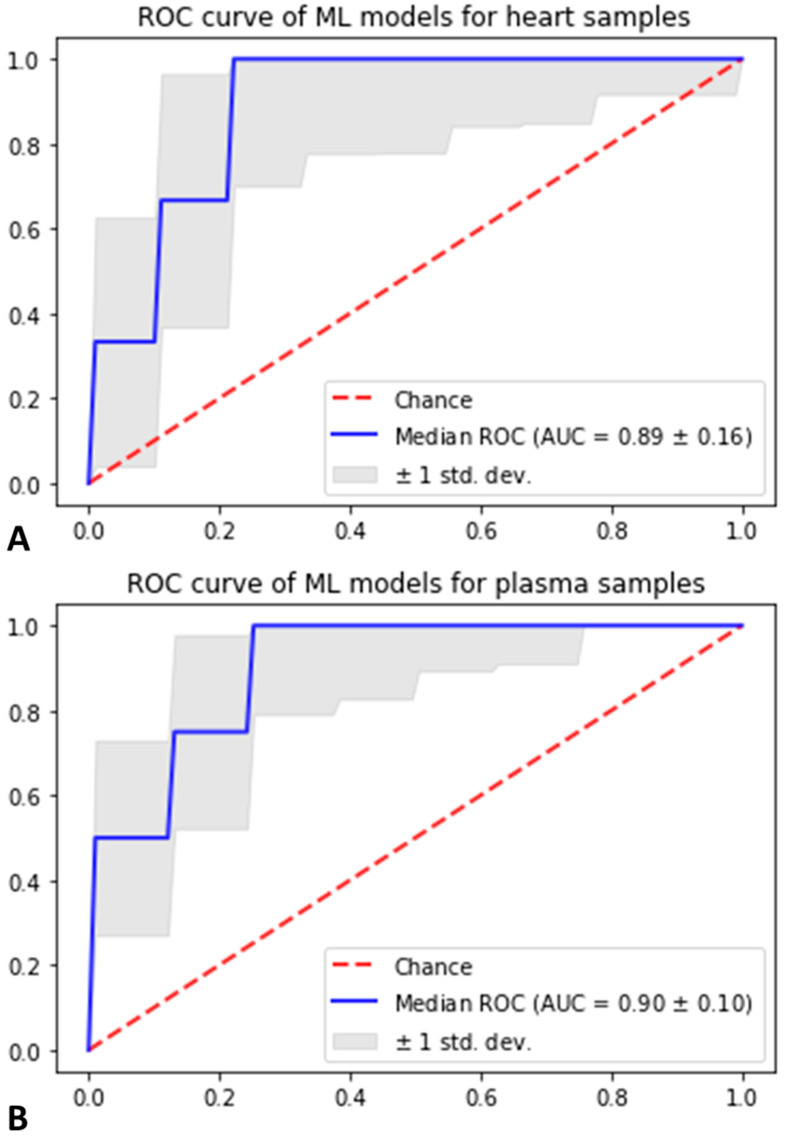
ROC AUC plots of median AUC achieved by machine learning models on heart (**A**) and plasma (**B**) samples. On each dataset, 30 different samples were split into training data and test data and used to train 30 different models, allowing better estimation of performance and selection of relevant variables.

**Figure 3 ijms-24-12000-f003:**
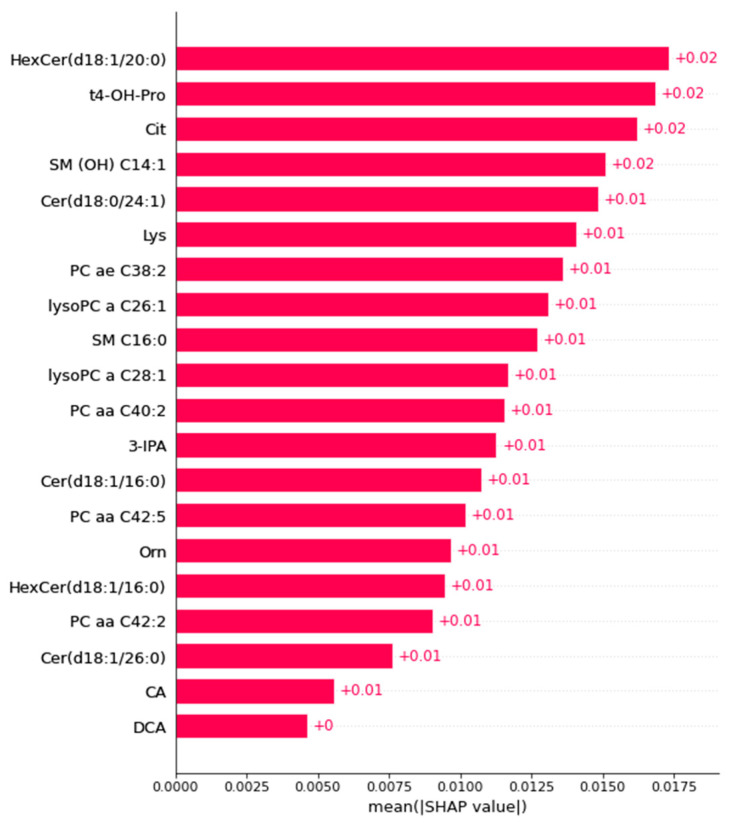
The average of all SHAP values of important plasma metabolites allowing us to distinguish between rats subjected to stroke and those not subjected to stroke. SHAP values were computed based on the predictions of the best model fitted on each of the 30 different train/test sampling of the plasma dataset. The most important metabolites have the highest SHAP values and are at the top and the least important have the lowest values and are at the bottom. CA: cholic acid, DCA: desoxycholic acid, 3-IPA: 3-indolepropionic acid. Cer: ceramides, HexCer: hexocylceramides. SM: sphingomyelin, PC: phosphatidylcholine, lysoPC: lysophosphatidylcholine, Cit: citrulline, Lys: lysine, Orn: ornithine.

**Figure 4 ijms-24-12000-f004:**
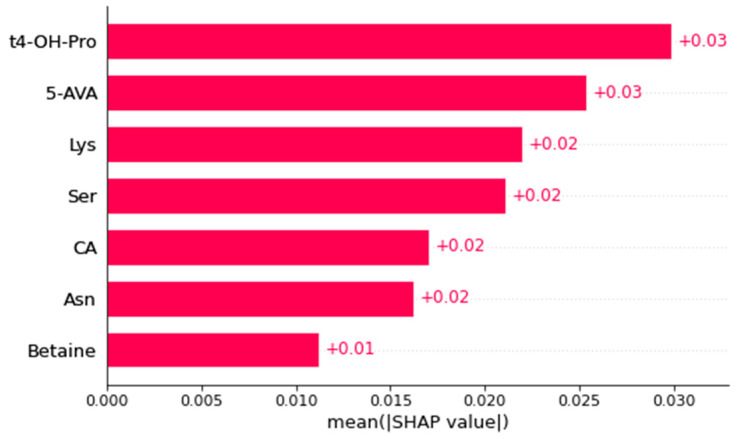
The average of all SHAP values of important heart metabolites allowing the discrimination of rats subjected to stroke or not. SHAP values were computed based on the predictions of the best model fitted on each of the 30 different train/test sampling of the heart dataset. The most important metabolites have the highest SHAP values and are at the top and the least important have the lowest values and are at the bottom. 5-AVA: acid 5-aminovaleric, CA: cholic acid, t4-OH-Pro: trans4-hydroxyproline, Lys: lysine, Ser: serine, Asn: asparagine.

**Figure 5 ijms-24-12000-f005:**
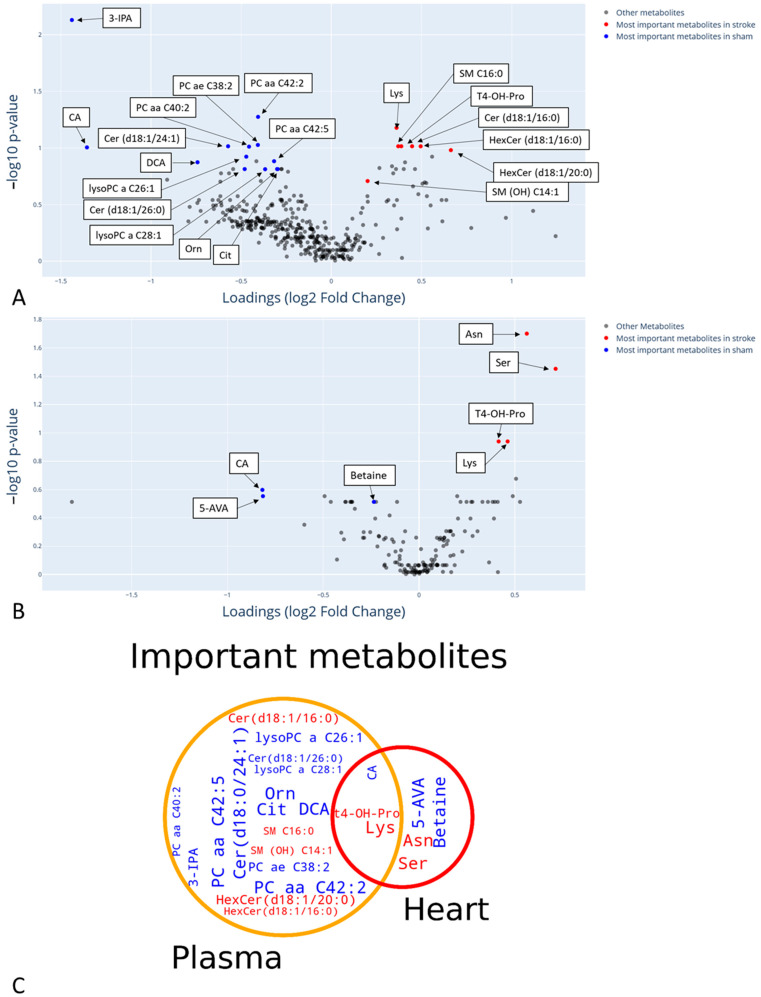
(**A**,**B**) Volcano plot of the metabolites found in plasma samples (**A**) and heart samples (**B**). Log2 fold change (stroke/sham) is in the *x*-axis and –log10 of the univariate p-value (after correction with Benjamini–Hochberg) in the *y*-axis. Important metabolites highlighted by the machine learning pipeline are in blue (if more abundant in the sham group) or in red (if more abundant in the stroke group). (**C**) Venn diagram of important metabolites altered by stroke based on plasma or heart samples. Important metabolites identified in plasma samples are inside the orange circle. Important metabolites identified in heart samples are inside the red circle. Metabolites in blue are at a higher concentration in the sham group while metabolites in red are at a higher concentration in the stroke group.

**Figure 6 ijms-24-12000-f006:**
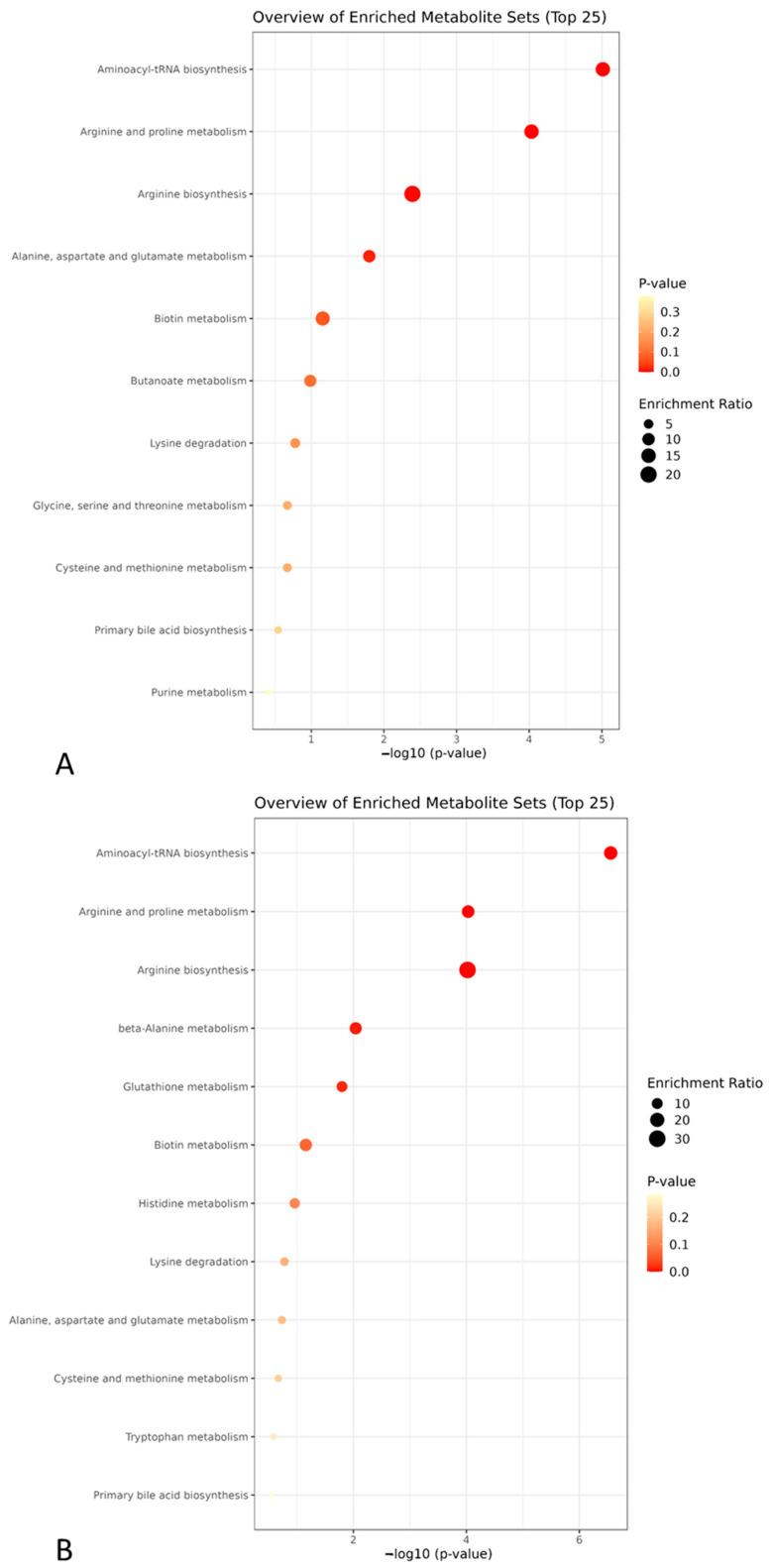
Metabolite set enrichment analysis for heart (**A**) and plasma (**B**) samples revealing significant metabolic pathways impacted by stroke (red dot).

**Table 1 ijms-24-12000-t001:** Descriptive statistics of the twenty metabolites contributing to the stroke prediction found in plasma samples. CA: cholic acid, DCA: desoxycholic acid, 3-IPA: 3-indolepropionic acid. Cer: ceramides, HexCer: hexocylceramides. SM: sphingomyelin, PC: phosphatidylcholine, lysoPC: lysophosphatidylcholine, Cit: citrulline, Lys: lysine, Orn: ornithine, Pro: proline, Gly: glycine, Hex2Cer: dihexocylceramides, HexCer: hexocylceramides, std: standard deviation.

	Sham				Stroke				
Feature	Mean	95% Confidence Interval (Mean)	Std	Range	Mean	95% Confidence Interval (Mean)	Std	Range	Effect Size (Cohen’s d)
HexCer(d18:1/20:0)	0.16	[0.14;0.19]	0.06	0.23	0.26	[0.2;0.32]	0.09	0.26	1.31
t4-OH-Pro	27.12	[24.63;29.61]	6.43	26.80	35.48	[31.46;39.51]	6.34	21.9	1.31
Cit	54.61	[50.30;58.91]	11.10	43	44.43	[37.24;51.63]	11.33	35.4	0.91
SM (OH) C14:1	0.56	[0.52;0.59]	0.09	0.35	0.64	[0.58;0.70]	0.10	0.32	0.89
Cer(d18:0/24:1)	0.10	[0.09;0.12]	0.03	0.11	0.07	[0.05;0.1]	0.04	0.14	1.10
Lys	352.79	[324.20;381.37]	73.72	302	453.25	[405.48;501.02]	75.19	226	1.35
PC ae C38:2	1.15	[1.06;1.25]	0.25	0.92	0.87	[0.78;0.96]	0.15	0.51	1.28
lysoPC a C26:1	1.65	[1.48;1.81]	0.42	1.55	1.19	[0.91;1.46]	0.43	1.49	1.08
SM C16:0	34.31	[31.38;37.24]	7.56	26.50	44.37	[39.07;49.67]	8.34	25.20	1.29
lysoPC a C28:1	0.94	[0.86;1.03]	0.21	0.81	0.73	[0.58;0.88]	0.24	0.76	0.97
PC aa C40:2	0.29	[0.26;0.31]	0.07	0.25	0.21	[0.17;0.25]	0.06	0.19	1.19
3-IPA	1.39	[1.08;1.71]	0.81	3.06	0.51	[0.15;0.87]	0.57	2.12	1.18
Cer(d18:1/16:0)	0.89	[0.81;0.97]	0.20	0.84	1.21	[1.00;1.42]	0.33	1.12	1.31
PC aa C42:5	0.23	[0.21;0.24]	0.04	0.16	0.18	[0.16;0.21]	0.04	0.13	1.18
Orn	115.46	[101;129.91]	37.28	162.6	93.74	[52.76;134.72]	64.50	246.30	0.46
HexCer(d18:1/16:0)	0.58	[0.52;0.65]	0.16	0.57	0.82	[0.65;1.00]	0.28	1.08	1.19
PC aa C42:2	0.22	[0.21;0.24]	0.04	0.18	0.17	[0.14;0.19]	0.04	0.15	1.35
Cer(d18:1/26:0)	0.47	[0.42;0.53]	0.15	0.58	0.34	[0.25;0.43]	0.14	0.52	0.92
CA	2.44	[1.69;3.19]	1.94	9.57	0.95	[0.01;1.90]	1.49	4.96	0.82
DCA	0.76	[0.46;1.05]	0.77	3.34	0.45	[−0.12;1.02]	0.90	3.17	0.38

**Table 2 ijms-24-12000-t002:** Descriptive statistics of the seven metabolites contributing to the stroke prediction found in heart samples. 5-AVA: acid 5-aminovaleric, CA: cholic acid, t4-OH-Pro: trans4-hydroxyproline, Lys: lysine, Ser: serine, Asn: asparagine, Pro: proline, Cit: citrulline, Gly: glycine, AAs: amino acids, std: standard deviation.

	Sham				Stroke				
Feature	Mean	95% Confidence Interval (Mean)	Std	Range	Mean	95% Confidence Interval (Mean)	Std	Range	Effect Size (Cohen’s d)
t4-OH-Pro	9.64	[8.99;10.29]	1.68	6.73	12.86	[10.5;15.22]	3.51	12.63	1.38
5-AVA	0.51	[0.4;0.62]	0.29	1.28	0.29	[0.15;0.43]	0.21	0.54	0.82
Lys	76.35	[69;83.69]	18.95	82.40	105.20	[87.58;122.82]	26.23	77.40	1.36
Ser	56.92	[51.03;62.81]	15.20	57.60	93.29	[68.06;118.53]	37.56	136.20	1.55
CA	0.08	[0.06;0.09]	0.05	0.18	0.04	[0;0.08]	0.06	0.20	0.65
Asn	39.43	[35.76;43.1]	9.47	34.70	58.23	[49.27;67.19]	13.34	46.90	1.76
Betaine	31.77	[28.96;34.58]	7.25	28.60	26.99	[19.84;34.14]	10.64	34.30	0.58

## Data Availability

The data presented in this study are available in the Appendix A (data after quality filtering).

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
