# Peer review of "Combined Metabolipidomic and Machine Learning Approach in a Rat Model of Stroke Reveals a Deleterious Impact of Brain Injury on Heart Metabolism"

_ijms, 2023, doi:10.3390/ijms241512000_

Round 1
Reviewer 1 Report
Dear authors, I read the work on reviewing with great interest. Addressing the evaluation of the various metabolites by means of machine learning makes sense to me and is contemporary. Basically, I would be happy to be able to read this work accepted publication, but some points still need to be addressed.
It should be made clearer which statistical analysis was carried out, since a not inconsiderable number of rats died during the procedure.
In addition, I would make it easier to understand both the actual experimental setup and the evaluation of a graphical representation.
In addition, further explanations should be added as to what the training set was.
Reviewer 2 Report
This is an excellent paper that investigates the metabolic changes in plasma and heart samples from rats after stroke or sham procedure, using targeted quantitative metabolomics and machine learning.
The paper is well-written, clear, and informative. The methods are rigorous and appropriate, and the results are robust and consistent.
The paper provides new insights into the metabolic alterations and pathways involved in stroke pathophysiology and recovery, as well as potential biomarkers and therapeutic targets.
The paper also discusses the limitations and future directions of the study, such as the need for validation in human samples and the exploration of other organs and time points.
I have only a few minor comments and suggestions for improvement:
- In the introduction, please provide some background information on the Biocrates metabolome kit, such as its advantages, disadvantages, and applications in previous studies.
- In the methods, please explain how the samples were randomized and allocated to different groups, and how the blinding was done.
- In the results, please provide some descriptive statistics (mean, standard deviation, range) for the metabolite concentrations in each group, as well as the effect sizes and confidence intervals for the comparisons between groups.
- In the discussion, please compare and contrast your findings with those of other studies that used similar or different methods or models of stroke.
- In the conclusion, please highlight the main contributions and implications of your study for stroke research and clinical practice.
I congratulate the authors on their excellent work.
Round 2
Reviewer 1 Report
Dear authors, dear editors,
From my point of view, the publication has clearly gained after the revision and I am in favour of publication.